# Nuclear Quantum Effects in the Ionic Dissociation Dynamics of HCl on the Water Ice Cluster

**DOI:** 10.3390/molecules30030442

**Published:** 2025-01-21

**Authors:** Tatsuhiro Murakami, Hinami Ueno, Yuya Kikuma, Toshiyuki Takayanagi

**Affiliations:** Department of Chemistry, Saitama University, Shimo-Okubo 255, Sakura-ku, Saitama City 338-8570, Saitama, Japan; h.ueno.687@ms.saitama-u.ac.jp (H.U.); y.kikuma.323@ms.saitama-u.ac.jp (Y.K.)

**Keywords:** nuclear quantum effects, ring-polymer molecular dynamics, ionic dissociation, hydrochloric acid, water ice cluster

## Abstract

Nuclear quantum effects play a significant role in the dissociation dynamics of HCl ions during collisions with the (H_2_O)_49_ ice cluster. These effects become particularly important when analyzing proton transfer, tunneling, and zero-point energy contributions during the dissociation process. In this study, we investigate the dissociation behavior of HCl when colliding with the (H_2_O)_49_ ice cluster, focusing on the influence of the nuclear quantum effects on the proton transfer mechanism, ionic dissociation rates, and subsequent solvation dynamics. Through a combination of classical molecular dynamics (MD) and ring-polymer molecular dynamics (RPMD) simulations, we explore how quantum fluctuations in the proton’s position alter the dissociation pathway of HCl. The inclusion of nuclear quantum effects reveals enhanced proton mobility, leading to differences in dissociation behavior compared to classical simulations. Our findings indicate that nuclear quantum effects significantly affect the dissociation dynamics, with the proton more readily transferring to the hydrogen-bond network in the (H_2_O)_49_ ice cluster. This study provides insights into the quantum mechanical nature of ionic dissociation in hydrogen-bonded systems and highlights the importance of incorporating nuclear quantum effects for accurate modeling of proton transfer processes in complex environments.

## 1. Introduction

The ionic dissociation of a strong acid, such as hydrochloric acid (HCl), which produces a proton (H^+^) and a chloride anion (Cl^−^), is a crucial process that facilitates subsequent chemical reactions. While it is well known that HCl undergoes near-complete dissociation in bulk water, its interaction with a single water molecule does not result in dissociation; instead, it forms the HCl(H_2_O) heterodimer [1,2]. To determine the specific number of water molecules required for HCl to dissociate into its ionic form, numerous theoretical [1,2,3,4,5,6,7,8,9] and experimental [10,11,12,13,14] studies have been conducted. HCl eventually dissociates in clusters containing four or more water molecules.

Understanding the mechanism of HCl ionic dissociation in water ice is of critical importance across various fields of chemistry, particularly in atmospheric chemistry [15] and astrochemistry [16]. In polar stratospheric clouds, HCl, which serves as a reservoir for non-ozone-depleting chlorine, is activated into Cl_2_ molecules [17]. Upon photodissociation, Cl_2_ produces chlorine atoms (Cl), which participate in reactions that deplete ozone [18]. Notice that the ionic dissociations have been experimentally observed under stratospheric conditions [19,20,21,22], indicating that the Cl^−^ anion could play an important role in the production of an active Cl atom. The dissociation mechanism of HCl in ice clusters in the interstellar medium has also attracted significant interest [16], as HCl has been detected and is considered one of the primary carriers of chlorine in the interstellar medium [23,24].

Various classical molecular dynamics simulations have been conducted to investigate the dissociation mechanism of HCl in water clusters using empirical force fields [25,26,27], and on-the-fly ab initio [28] and quantum mechanics molecular mechanics (QM/MM) calculations [29]. Although the dynamic results indicate the formation of a contact ion pair (CIP) and a solvent-separated ion pair (SSIP), nuclear quantum effects were not accounted for in the simulations. This suggests that the proton transfer rate may be underestimated in classical molecular dynamics simulations, as nuclear quantum effects play a significant role in chemical reactions involving protons. From a static viewpoint, based on probability density analyses using a path-integral technique, classical and quantum fluctuating structures of the HCl(H_2_O)_4_ clusters were significantly different [30,31,32,33], highlighting the critical role of nuclear quantum effects in proton transfer and the hydrogen-bond network within the clusters.

In this study, ring-polymer molecular dynamics (RPMD) [34,35,36,37], a technique that captures the quantum mechanical behavior of nuclei by representing them as cyclic beads, was performed alongside classical MD simulations, where the cyclic bead count is set to one, effectively neglecting nuclear quantum effects. These approaches were used to investigate the influence of nuclear quantum effects on ionic dissociation dynamics. Specifically, head-on collision simulations between HCl and the amorphous ice cluster (H_2_O)_49_ [38] at 250 K, representing a simple model of polar stratospheric conditions [39], were performed. As previously mentioned, HCl undergoes ionic dissociation in the presence of four or more water molecules. The ice cluster, consisting of 49 water molecules and representing amorphous ice, as developed by Sameera and coworkers [38], is sufficient for investigating the mechanisms of HCl ionic dissociation through head-on collision simulations. Details of the calculations for potential energies and their gradients used in the RPMD and classical MD simulations are provided in the Methodology section. The total number of trajectories for both classical MD and RPMD simulations is chosen to 100. The initial amorphous water cluster (H_2_O)_49_ structure before thermalization was taken from Ref. [38] and is shown in Appendix A. The Cartesian coordinates in xyz format are provided in Appendix A. The detailed procedures for the collision simulations are described in the Methodology section. In this paper, we report the survival lifetime of the neutral HCl + (H_2_O)_49_ structure, along with the time-dependent evolution of the CIP (H_2_O)_48_H_3_O^+^Cl^−^ and the SSIP H_3_O^+^(H_2_O)_48_Cl^−^ structures, as depicted in Figure 1. Additionally, isotope effects were also examined through DCl + (H_2_O)_49_ collision simulations to provide a more quantitative analysis of the nuclear quantum effects. Similar to the HCl + (H_2_O)_49_ simulations, a total of 100 trajectory simulations were conducted.

## 2. Results and Discussion

Figure 2a–d display the time-evolved populations of the neutral state, CIP and SSIP for HCl + (H_2_O)_49_ using classical MD and RPMD, and for DCl + (H_2_O)_49_ using classical MD and RPMD, respectively. Each red line represents the single-exponential function to obtain the lifetime of the neutral states. The timing of the first collision between HCl and (H_2_O)_49_ is defined as $t_{c}=0\;$ps. As shown in Figure 2a,b, the neutral survival time ($\tau_{N}$) in RPMD ($\tau_{N}=6.27$ ps) is significantly shorter than that in classical MD ($\tau_{N}=24.36$ ps), indicating that the ionic dissociation rate of HCl on the water cluster is underestimated in classical MD simulations. Similar to the HCl + (H_2_O)_49_ collisions, the lifetime of the DCl + (H_2_O)_49_ reaction in RPMD ($\tau_{N}=9.41$ ps) is quite shorter than that in classical MD ($\tau_{N}=28.99$ ps), as depicted in Figure 2c,d. The results suggest that a proper description of quantized vibrational energies and quantum fluctuations is essential for appropriately evaluating the ionic dissociation process and describing the proton (deuteron) transfer barriers. The exponential curves represent that the decay rates of HCl in classical MD and RPMD are approximately 1.2 and 1.5 times faster, respectively, than those of DCl in the corresponding simulations. Compared to classical MD, RPMD exhibits a more pronounced isotope effect, reflecting the over-the-barrier dynamics leading to the ionically dissociated form.

A comparison between Figure 2a and Figure 2b, as well as between Figure 2c and Figure 2d, reveals that the population of SSIP in the RPMD scheme is significantly larger than in classical MD. This indicates that the proton transfer in the hydrogen-bond network of the water cluster occurs more frequently after the formation of CIP in the RPMD simulations. Figure 3 shows the distribution of transient lifetimes of hydronium ion (H_3_O^+^) following ionic dissociation. In other words, the lifetimes represent the duration for which H_3_O^+^ remains intact, constructed from the specific atoms involved. The transient lifetimes in RPMD are notably shorter than those in classical MD, indicating that proton transfers occur more readily due to quantum effects, as expected. The lifetimes of HCl + (H_2_O)_49_ and DCl + (H_2_O)_49_ collisions are similar because the hydrogens in the water cluster were not deuterated. In this study, the Eigen and Zundel forms were not distinguished in the dynamics simulations, reflecting the frequent occurrence of proton transfers within the hydrogen-bond network.

To investigate the location where the proton transfer occurs, the time-evolved normalized probability densities of the internuclear distance between Cl and the transient proton are illustrated for HCl + (H_2_O)_49_ using classical MD and RPMD, and for DCl + (H_2_O)_49_ using classical MD and RPMD in Figure 4a, Figure 4b, Figure 4c, and Figure 4d, respectively. The transient proton is defined as the hydrogen atom with the third-largest OH distance in H_3_O^+^ after ionic dissociation. For the neutral case, the bond lengths of HCl are described as the internuclear distance between Cl and the proton in Figure 4. At $t_{c}=0\;$ps, when HCl first collides with the water cluster, the probability densities are similar to those of isolated HCl, as determined using the path-integral scheme at 250 K, and are shown in Appendix A. As time progresses, the internuclear distances become widely distributed, suggesting that various SSIP structures are formed through numerous proton transfers. Notably, the probability densities at larger distances in RPMD are greater than those in classical MD, as shown in Figure 4.

Figure 5 shows the time-evolution of the coordination number of the Cl^−^ anion to understand the solvation structures. In this study, the coordination number CN is defined using the following equations [40]:(1)$$CN=\sum_{i\in H}c_{i},$$(2)$$c_{i}=\frac{{1-\left(\frac{d_{i}}{R_{0}}\right)}^{6}}{{1-\left(\frac{d_{i}}{R_{0}}\right)}^{12}},$$
where $c_{i}$ is a switching function. $d_{i}$ represents the distance between the Cl atom and the *i*-th hydrogen atom, while $R_{0}$ denotes the cutoff distance, set to 2.4 Å in this study. In the RPMD case, which exhibits larger populations of the CIP and SSIP structures, the coordination number increases significantly to approximately 4, with its distribution notably higher than that in the classical case, as time progresses. This suggests that the Cl^−^ anion interacts with four hydrogen atoms of the water cluster through structural rearrangements involving H_3_O^+^(H_2_O)_48_. The coordination number for the SSIP state in this study aligns with the minimum energy structures of HCl(H_2_O)*_n_* with *n* = 9–15 and 21 [8]. Furthermore, the population of CN≈3 remains minimal at each time step, indicating relatively rapid structural changes. Moreover, the populations of the SSIP state, especially in the RPMD case, are relatively more localized compared to those of the CIP state, and the peak positions of the coordination number for the SSIP are closer to 4 than those for the CIP (see Appendix A). Notice that the coordination number derived from the above equations with the cutoff distance for the neutral state is slightly overestimated because the covalent bond of HCl (See Appendix A) is relatively shorter than the ionic interaction.

## 3. Methodology

### 3.1. Potential Energy Calculations

On-the-fly RPMD and classical MD simulations were performed for HCl + (H_2_O)_49_ and DCl + (H_2_O)_49_ collision simulations in this study. The potential energies and their gradients were obtained using the GFN2-xTB method developed by Grimme and coworkers [41,42,43]. The GFN2-xTB method, a semiempirical quantum chemical approach, is capable of effectively representing chemical bond cleavage and formation, unlike empirical force-field methods. While the potential energy calculations using the GFN2-xTB method for structures near equilibrium are consistent with those obtained from first-principle quantum chemical methods, the GFN2-xTB method exhibits lower accuracy for configurations far from equilibrium, such as those at the dissociation limit. Therefore, the empirical chemical element parameters for hydrogen and oxygen atoms of the GFN2-xTB method were optimized to fit the first-principle DFT energies and their gradient at the 400 configuration points, which were obtained from the preliminary trajectory calculations for the HCl(H_2_O)_5_ system performed at the B3LYP/6-311++G(d,p) level using Gaussian09 [44]. The root-mean-square-errors (RMSEs) for both energy and gradient showed slight improvement (See Appendix A). The optimized parameters are listed in Appendix A and were used to perform the on-the-fly dynamics simulations for the HCl + (H_2_O)_49_ and DCl + (H_2_O)_49_ systems. The optimization was performed using a genetic algorithm [45]. Dynamics simulations with the optimized GFN2-xTB calculations have already been successfully applied to various systems [46,47,48].

### 3.2. Procedure for Molecular Dynamics

An amorphous water cluster consisting of 49 water molecules (H_2_O)_49_, developed by Sameera and coworkers [38], was selected as the initial structure for thermalization at 250 K. In this study, dynamics simulations were carried out for an isolated system comprising an HCl molecule and a water ice cluster. A total of 100 initial coordinates and momenta, where the HCl(DCl) and (H_2_O)_49_ fragments do not interact, were generated using path-integral molecular dynamics (PIMD) simulations with the Nosé-Hoover thermostat at 250 K. PIMD provides the quantum Boltzmann distribution, effectively incorporating nuclear quantum effects, including discretized vibrational energies, using an appropriate number of beads (*N*_bead_ = 16 in this study). The equations of motion for the ring-polymer Hamiltonian [49] were solved using the velocity−Verlet method with a time step of Δ*t* = 0.20 fs, totaling 10^4^−10^5^ simulation steps. The 100 initial configurations of coordinates and momenta for the real-time dynamics simulations were sampled at intervals of 100 steps from the 2 × 10^4^ to 3 × 10^4^ steps of the PIMD simulations. After completing the PIMD simulations, the momentum vectors of fragments in the initial conditions were rotated based on the relationship between the Cartesian and the spherical polar coordinates to ensure collisions. Further details of the rotational procedure are provided in the Supporting Information of Ref. [49]. Subsequently, RPMD simulations were conducted, extending the PIMD method to enable real-time dynamics, which is particularly effective in capturing nuclear quantum effects such as zero-point energy and tunneling [49,50,51,52,53,54,55,56,57,58,59,60]. Note that while the accuracy of reaction rate coefficients obtained from RPMD simulations has been well demonstrated [34,35,61,62,63,64], careful attention should be given to the integration of the equations of motion when studying vibrational and absorption spectra [65,66], due to potential issues arising from the chain resonance problem [55,67]. As an alternative approach to addressing the chain resonance problem, the Brownian Chain Molecular Dynamics (BCMD) method, which incorporates randomized bead motions during the dynamics, has been recently developed by Shiga [68]. In this study, the RPMD trajectories were propagated by integrating the equations of motion of the ring-polymer Hamiltonian without a thermostat, employing a time step of Δ*t* = 0.20 fs. The first collision event ($t_{c}=0\;$ps) was defined as the moment when the minimum centroid distance between the hydrogen atom of HCl and any oxygen atom of water first becomes less than 2.0 Å. The dynamics simulations after the first collision were carried out for a minimum duration of 6 ps. Classical MD simulations were also performed for comparison with the RPMD results, following a similar procedure except that the number of beads was fixed at one. Notice that the PIMD simulations with *N*_bead_ = 1 were performed, and 100 initial conditions were extracted at intervals of 1000 steps between 2 × 10^4^ and 1.2 × 10^5^ steps of the PIMD simulations. The open-source code PIMD.ver.2.6.0. [69] was utilized to conduct all PIMD, RPMD, and classical MD calculations.

## 4. Conclusions

We investigated the ionic dissociation reactions of HCl and DCl in a water ice cluster using classical MD and RPMD collision simulations. The survival lifetimes of the neutral states for both HCl and DCl in the RPMD scheme are significantly shorter than those in classical MD, highlighting the crucial role of nuclear quantum effects in the dissociation into H^+^ and Cl^−^. In RPMD, the ionic dissociation rate for DCl is relatively smaller than that for HCl, suggesting that quantized vibrational energies and quantum fluctuations are key factors in the ionic dissociation process. Following ionic dissociation, the coordination number of Cl increases to approximately 4 due to structural rearrangements in the water cluster. The transient lifetimes of H_3_O^+^ following ionic dissociation, as observed in RPMD, are considerably shorter than those in classical MD. This indicates that quantum effects significantly facilitate proton transfers, enabling them to occur more readily. The transient lifetimes for HCl and DCl reactions are similar because the hydrogen atoms in the water cluster were not replaced with deuterium. In this study, we focused on nuclear quantum effects in the ionic dissociation mechanism by conducting RPMD simulations at 250 K. Our simulations demonstrated that nuclear quantum effects play a crucial role in ionic dissociation, proton transfer rates, and the dynamical changes in the coordination number of Cl. It is worth noting that the temperature dependence of the ionic dissociation rate in the low-temperature range remains a topic of debate [16]. This aspect, which lies beyond the scope of the present study, will be addressed in a future report.

## Figures and Tables

**Figure 1 molecules-30-00442-f001:**
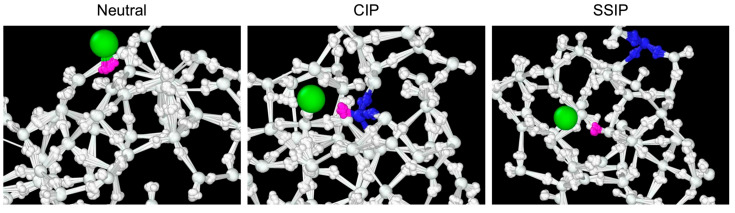
Ring-polymer representations of the neutral, undissociated HCl structure (**left panel**), contact ion pair (CIP, **middle panel**) and solvent-separated ion pair (SSIP, **right panel**). The green, magenta, blue and white colors correspond to Cl^−^, the original proton H^+^ from HCl, H_3_O^+^ and the remaining H_2_O cluster, respectively.

**Figure 2 molecules-30-00442-f002:**
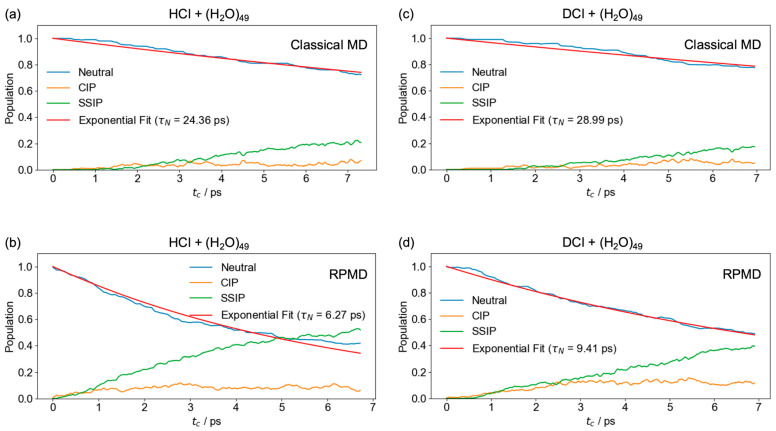
Time-evolved populations of the neutral state (blue line), CIP (orange line) and SSIP (green line) for HCl + (H_2_O)_49_ using (**a**) classical MD and (**b**) RPMD, and for DCl + (H_2_O)_49_ using (**c**) classical MD and (**d**) RPMD. The red line represents the exponential decay function for the neutral state. The $t_{c}=0$ ps indicates the moment of the first collision between HCl and (H_2_O)_49_.

**Figure 3 molecules-30-00442-f003:**
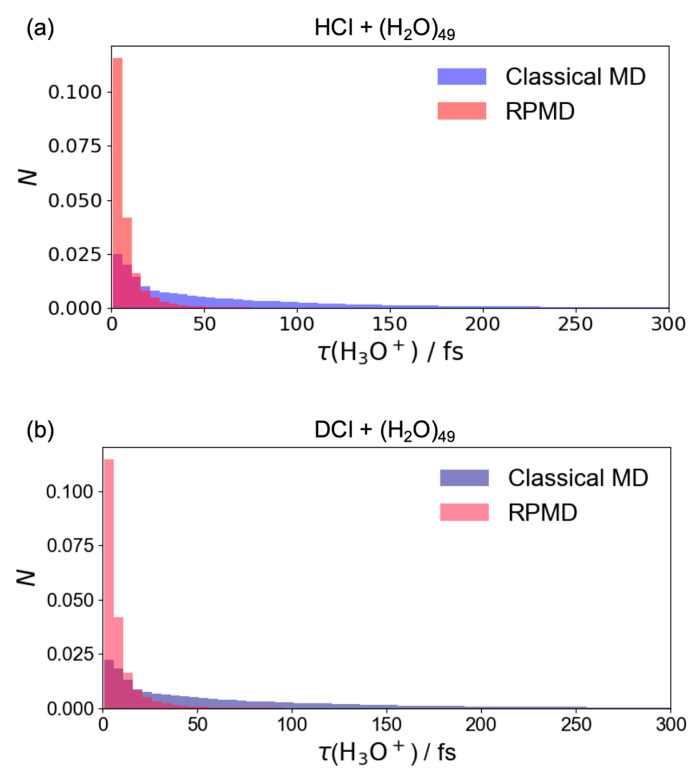
Transient lifetimes of H_3_O^+^ after ionic dissociation for classical MD (blue bars) and RPMD (red bars) in terms of (**a**) HCl + (H_2_O)_49_ and (**b**) DCl + (H_2_O)_49_ reactions.

**Figure 4 molecules-30-00442-f004:**
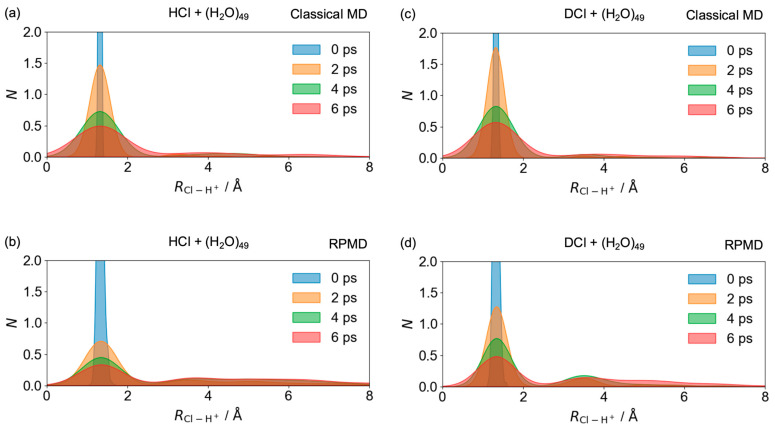
Snapshots of the internuclear distance between the Cl and the transient proton within hydrogen-bond network at $t_{c}=$0 (blue), 2 (orange), 4 (green) and 6 ps (red), for HCl + (H_2_O)_49_ using (**a**) classical MD and (**b**) RPMD, and for DCl + (H_2_O)_49_ using (**c**) classical MD and (**d**) RPMD. Notice that these plots include the HCl distance of the neutral state.

**Figure 5 molecules-30-00442-f005:**
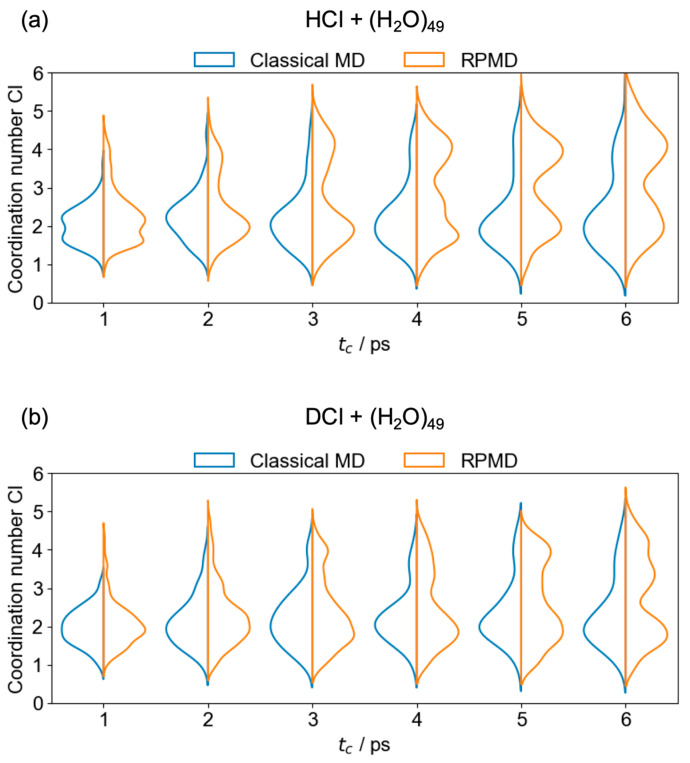
Time-evolved violin plots in terms of the coordination number of Cl in classical MD (blue) and RPMD (orange) for (**a**) HCl + (H_2_O)_49_ and (**b**) DCl + (H_2_O)_49._

## Data Availability

Data are contained within the article.

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
