# Peer review of "Nuclear Quantum Effects in the Ionic Dissociation Dynamics of HCl on the Water Ice Cluster"

_molecules, 2025, doi:10.3390/molecules30030442_

Round 1
Reviewer 1 Report
Comments and Suggestions for Authors
In this manuscript, the author presented research on the influence of nuclear quantum effect in dissociation of HCl in water ice. Ring-polymer molecular dynamics simulations revealed significant shortening in lifetime for neutral species, and several details such as Cl-H distance and coordination number are presented. I believe this is a solid work and recommend publication, and I’ll be grateful if the authors can resolve some of my minor concerns:
Minor concerns:
1. There’s a significant difference in proton transfer rate from MD and RPMD. Is there an experimental record that can support the result of RPMD?
2. In Line 211, why there are '104-105 steps'? I’m confused that for all 100 trajectories the number of the steps should be the same, instead of falling into a range.
3. 100 initial conditions for both HCl and water cluster can be generated from PIMD. Then the authors rotated the initial coordinates & momentum of HCl and water cluster obtained to ensure collision. How is this achieved? Is there a general rule for this, or is it done all by hand?
Reviewer 2 Report
Comments and Suggestions for Authors
The manuscript addresses the critical role of nuclear quantum effects in the dissociation dynamics of HCl on water ice clusters. The authors utilized RPMD alongside MD and emphasized that nuclear quantum effects play a significant role in HCl dissociation. I would like to recommend this manuscript for publication in Molecules after addressing the following issues:
1. The rationale for selecting (H2O)49 as the ice cluster size could be better justified. Perhaps it is representative of polar stratospheric clouds or interstellar medium conditions, but this needs to be clarified.
2. To the best of my knowledge, classical MD cannot accurately capture bond breaking and bond forming. If this is correct, the authors’ interpretation may be misleading.
3. Discuss potential sources of error in the RPMD and classical MD simulations, such as limitations of the GFN2-xTB method or the bead representation in RPMD.
4. In the computational details, what is the size of the simulation box?
5. The comparison between isotope effects in DCl and HCl could delve further into the implications of these findings for hydrogen-bond network dynamics.
Reviewer 3 Report
Comments and Suggestions for Authors
The paper offers insightful information about nuclear quantum effects on ionic dissociation kinetics and is well structured. Resolving the remarks would improve readability and clarity.
1. Explain why (H2O)49 was chosen as the ice cluster size and how it relates to actual circumstances like interplanetary or stratospheric environments.
2. Make sure legends clearly distinguish between RPMD and traditional MD results to improve figure clarity. To make the overlapping data points in Figure 4 easier to read, enlarge or divide the insets.
3. To increase clarity and guarantee consistency throughout the text, standardize the usage of phrases like "neutral state" and acronyms (such as CIP and SSIP).
Round 2
Reviewer 2 Report
Comments and Suggestions for Authors
After reading the authors' response. I would like to recommend the manuscript to be published in Molecules journal